# Peri-Operative Antimicrobial Prophylaxis Modulates CD4^+^ Lymphocyte Immunophenotype *Ex Vivo* in High-Risk Patients Undergoing Major Elective Surgery—A Preliminary Observational Study

**DOI:** 10.3390/antibiotics14101026

**Published:** 2025-10-14

**Authors:** Susi Paketci, Jack Williams, Walter Pisciotta, Richard Loye, Alessia V. Waller, Rahila Haque, David Brealey, Mervyn Singer, John Whittle, Ramani Moonesinghe, Nishkantha Arulkumaran, Timothy Arthur Chandos Snow

**Affiliations:** 1Bloomsbury Institute of Intensive Care Medicine, University College London, London WC1E 6BT, UK; susi.paketci@nhs.net (S.P.); jack.williams13@nhs.net (J.W.); w.pisciotta@ucl.ac.uk (W.P.); richard.loye.20@ucl.ac.uk (R.L.); alessia.waller1@nhs.net (A.V.W.); scv25gdu@uea.ac.uk (R.H.); d.brealey@nhs.net (D.B.); m.singer@ucl.ac.uk (M.S.); nisharulkumaran@doctors.org.uk (N.A.); 2NIHR UCLH Biomedical Research Centre, University College London Hospitals NHS Foundation Trust, London NW1 2BU, UK; ramani.moonesinghe@nhs.net; 3Centre for Anaesthesia, Critical Care & Pain Medicine, University College London, London WC1E 6BT, UK; j.whittle@ucl.ac.uk

**Keywords:** anti-bacterial agents, beta-lactams, general surgery, immune function, intensive care unit, lymphocyte, monocyte, post-anaesthetic care unit, post-operative infection

## Abstract

**Background**: Post-operative infections are a significant cause of morbidity in patients undergoing major elective surgery. Peri-operative antibiotics are used to reduce the risk of infection. Several antibiotics modulate the host immune response. **Objectives**: Our objective was to determine the *ex vivo* immunomodulatory properties of commonly used antibiotics (amoxicillin, cefuroxime, metronidazole, or combined cefuroxime–metronidazole) on monocyte and lymphocyte phenotypes in patients undergoing major elective surgery. **Methods**: We performed a prospective cohort study of patients aged ≥18 years admitted to the post-anaesthetic care unit following major elective non-cardiac surgery. Peripheral blood mononuclear cells isolated immediately after surgery were incubated with antibiotics with or without a monocyte (heat-killed *E. coli*) or lymphocyte (CD3/CD28 beads) stimulus *ex vivo*. Immune cell phenotype was characterised using flow cytometry. **Results**: Twenty-eight patients were included. All antibiotics tested were associated with a reduction in T-cell viability, and changes to monocytes were minimal. Among CD4^+^ and CD8^+^ lymphocytes, cefuroxime increased IFN-γ (at low and high doses) and increased CD4^+^ lymphocyte IL-2 and IL-2R at higher doses. Among CD4^+^ lymphocytes, at both doses, cefuroxime increased %T_h1_ population, with a parallel decrease in %T_h2_, %T_h17_, IL-17A, FOX-P3, and T-bet. Among the T_h1_ sub-population, changes were seen at higher cefuroxime doses, including increased viability and PD-1, and a decrease in FAS, IFN-γ and CD28, and IL-7R expression. **Conclusions**: The choice of antibiotics directly impacts immune function following major surgery, with cefuroxime associated with *ex vivo* immunomodulatory effects on CD4^+^ lymphocytes. The functional implications on the development of subsequent post-operative infectious complications and long-term cancer-free survival require further investigation.

## 1. Introduction

Post-operative infections (encompassing pneumonia, surgical-site infections, etc.) are a significant cause of morbidity. The reported incidence varies between studies, depending on the definition and reporting of infections, the patient cohort, and type of surgery; ranging from 9% in a global survey of all surgical specialities [1], to 40% of patients undergoing major non-cardiac surgery at tertiary referral specialist centres [2]. Surgery activates the immune system in response to physical injury to tissues (‘sterile inflammation’), which shares many similarities with the immune changes seen in sepsis [3]. An immunophenotype consistent with immunosuppression is commonplace in patients who develop post-operative infections and those with sepsis who develop secondary infections [4,5]. Several of the immune responses following surgery, such as a reduction in monocyte HLA-DR and persistent lymphopenia, are associated with development of subsequent infections [6,7,8] and are similar to the changes seen in patients who die with subsequent infections following sepsis [9].

The conventional approach to reducing the risk of post-operative infections is the liberal use of antibiotics [10]. However, there has been considerable effort to reduce the excessive use of antibiotics due to the increasing prevalence of antimicrobial resistance; guidelines recommend only administrating in cases of clean surgery with prosthesis insertion, or if surgical contamination occurs [11]. Beta-lactams (penicillins and cephalosporins) and nitroimidazoles are the most frequently used classes in a recent meta-analysis of antimicrobial prophylaxis regimens [12] (Appendix A), with amoxicillin, cefuroxime, and metronidazole currently recommended by UK guidelines [13]. Cefuroxime and metronidazole may be better in preventing post-operative infections compared to other antibiotics, related to their effect on different antimicrobial spectra [14].

In addition to concerns regarding antimicrobial resistance, understanding and awareness of antibiotic side-effects are increasing. Pre-clinical data demonstrates inadvertent effects of antibiotics on the immune system [15]. However, it is unclear if this affects patient outcomes, including the risk of late infections [16].

Our primary objective was to evaluate the *ex vivo* effect of amoxicillin, cefuroxime, metronidazole, and combined cefuroxime–metronidazole on immune cell phenotype in patients undergoing major surgery.

## 2. Results

### 2.1. Study Participants

Twenty-eight patients were included in this study (antibiotic characterisation cohort). An initial analysis of changes to immune cell phenotype was conducted in 12 patients (immunophenotyping cohort). Based on these data, a focused and in-depth characterisation of immune cell phenotype was conducted in another 16 patients (lymphocyte characterisation cohort). Patients had a median age of 64 (54–75) and 54% were male. 16/28 (57%) underwent upper gastrointestinal, 5/28 (18%) maxillofacial, 5/28 (18%) gynaecological, and 2/28 (7%) lower GI surgery. Most patients 27/28 (96%) underwent surgery for cancer resection, of whom 17/28 (61%) had received neoadjuvant chemotherapy. All patients received peri-operative antimicrobial prophylaxis; 17/28 (61%) received cefuroxime and metronidazole, 5/28 (18%) received co-amoxiclav, and 2/28 (7%) ciprofloxacin and clindamycin. Median duration of prophylactic antibiotic administration was 1(1–1) day. (Table 1 and Appendix A).

### 2.2. Effects of Antibiotics on Post-Operative PBMCs in the Immunophenotyping Cohort

In CD4^+^ and CD8^+^ lymphocytes, high-dose amoxicillin increased IFN-γ concentration. (Appendix A) Additionally, in CD8^+^ lymphocytes, high-dose amoxicillin increased CD28 expression (Appendix A) with decreased CTLA-4 expression (Appendix A) and cell viability (Appendix A). CD4^+^ and CD8^+^ lymphocyte viability was reduced by metronidazole at low (Appendix A) and high doses (Appendix A). Amoxicillin and metronidazole had no effect on monocyte phenotype. (Appendix A).

In CD4^+^ and CD8^+^ lymphocytes, cefuroxime increased IFN-γ concentration (Figure 1(a.i.,a.ii.,c.ii.,c.iii.), and Appendix A) and decreased viability (Figure 1(a.i.,a.ii.,c.i.), and Appendix A) at both doses. There were additional effects at a high dose, including an increase in CD4^+^ lymphocyte IL-2 concentration (Figure 1(b.i.,b.iii.), and Appendix A) and IL-2R (Figure 1(b.ii.,b.iii.), and Appendix A), and decreased CD8 lymphocyte CTLA-4 (Appendix A) and increased IL-2R expression (Appendix A). Cefuroxime had no effect on monocyte phenotype. (Appendix A).

Cefuroxime is often co-administered with metronidazole. When combined with metronidazole, the immunomodulatory effects were similar to that of cefuroxime alone. (Appendix A) This data suggested cefuroxime may have an immunomodulatory role on lymphocyte function and/or differentiation. We therefore proceeded to evaluate detailed T-cell subset phenotype changes induced by cefuroxime.

### 2.3. Cefuroxime Promotes CD4^+^ Lymphocyte Differentiation Towards a T_h1_ Phenotype in the Lymphocyte Characterisation Cohort

IL-17A was suppressed at both doses in CD4^+^ and CD8^+^ lymphocytes. (Figure 2(a.i.,a.ii.), Appendix A. CCR4, T-bet and Fox-P3 were reduced in CD4^+^ lymphocytes. (Figure 2(a.i.,a.ii.,b.ii.), Appendix A). Among CD4^+^ and CD8^+^ lymphocytes, proportions of T_h1_ (Figure 2(b.i.,b.ii.), Appendix A) and T_c1_ cell populations (Appendix A) increased with a parallel decrease in T_h2_, T_h17_, and T_c17_ cell populations at both low and high dose cefuroxime (Figure 2(a.i.,a.ii.), Appendix A). As cefuroxime increased the proportion of T_h1_ subsets and CD4^+^ lymphocyte IFN-γ, we further characterised cefuroxime-related changes to T_h1_ lymphocytes.

### 2.4. Cefuroxime Has an Immunomodulatory Effect on T_h1_ Subset Function in the Lymphocyte Characterisation Cohort

At a high dose, cefuroxime was associated with reduced activation of T_h1_ lymphocytes with reduced CD28, IFN-γ, and Fas, and a trend towards increased expression of PD-1 and reduced expression of IL-7R, although changes did not reach statistical significance. (Figure 3(a.ii.,b.i.,b.ii.,b.iii.), Appendix A) This reduced activation-associated cell death as viability was increased (Figure 3(b.iii.) and Appendix A). No immunomodulatory effect was seen with low dose cefuroxime. (Figure 3(a.i.) and Appendix A).

Cefuroxime was associated with lower levels of supernatant IL-10 at 72 h in unstimulated cells, with no differences at earlier timepoints. (Appendix A) There were no differences in IFN-γ at any time point. (Appendix A).

### 2.5. Effect of Stimulus on Post-Operative Immune Cell Phenotype in the Immunophenotyping Cohort

Following 24 h incubation with heat-killed *E. coli*, there was an increase in intracellular cytokine expression (IL-1β and IL-6), a decrease in chemokine receptor expression (CXCR4), cell viability, and population percentage. Markers associated with antigen presentation demonstrated an immunosuppressive phenotype (increased CD80, decreased CD86, and a trend towards suppressed HLA-DR). (Appendix A).

After 72 h incubation with CD3-28 beads, CD4^+^ lymphocytes demonstrated an increase in cytokine expression (IFN-γ) and markers of differentiation (IL-2 expression), with a decrease in co-stimulatory receptor (CD28) and proliferation (IL-7R). (Appendix A) CD8^+^ lymphocytes showed an increase in cytokine expression (IFN-γ and IL-10) and markers of differentiation (IL-2 expression), with a decrease in co-stimulatory receptor (CD28) proliferation (IL-7R) and anergy (PD-1). (Appendix A).

### 2.6. Effects of Antibiotics on Stimulated Post-Operative Cells in the Immunophenotyping Cohort

Amoxicillin had no effect on stimulated monocytes or lymphocytes. (Appendix A). Metronidazole also had no effect on stimulated monocyte phenotype. (Appendix A). In stimulated CD8^+^ lymphocytes, metronidazole decreased cell viability at a low dose (and high dose in univariate analysis), with no changes in CD4^+^ phenotype. (Appendix A).

Cefuroxime had no effect on stimulated monocyte immunophenotype, (Appendix A) but in stimulated CD4^+^ and CD8^+^ lymphocytes, cefuroxime increased IFN-γ concentration and decreased viability at a high dose only. (Appendix A) In stimulated lymphocytes, no additional effects were seen on lymphocyte differentiation (Appendix A) or on T_h1_ subset function. (Appendix A).

Combined cefuroxime–metronidazole had no effect on stimulated monocytes, but in stimulated CD4^+^ and CD8^+^ lymphocytes, it caused a decrease in viability and an increase in IFN-γ concentration at a high dose only. (Appendix A).

There were no differences in either released IL-10 or IFN-γ in stimulated PBMCs. (Appendix A).

### 2.7. Post-Operative Infections

We assessed whether the *ex vivo* effects of cefuroxime were associated with differences in the incidence of post-operative infections. Analysis was performed on our previously published clinical cohort, which from which the current dataset was derived [8], including 83 patients, 42 (51%) of whom were diagnosed with a post-operative infection, and occurring on day 3 (2–4). Patients either received cefuroxime (n = 53, alone or combined with other antibiotics), or any other antibiotic (n = 30, alone or in combination).

The other antibiotic cohort represented patients undergoing maxillofacial and gynaecological surgery, with a higher proportion of female patients in this group (67% vs. 32%), and fewer patients had received neo-adjuvant chemotherapy (33% vs. 62%) (Appendix A). Among patients receiving cefuroxime, 27 patients (51%) developed an infection, compared to 15 patients (50%) receiving other antibiotics (*p* = 0.831).

## 3. Discussion

Surgery induces several changes to the immune system [17,18,19]. We have previously described specific changes associated with post-operative infections including the following: elevated monocyte cell count, reduced monocyte chemokine receptor expression (CXCR4), and an increase in CD4^+^ lymphocyte IL-7R expression [8]. Here, we provide *ex vivo* evidence that the choice of antimicrobial prophylaxis may influence these changes.

In summary, we found that all antibiotics tested were associated with a reduction in lymphocyte viability, and changes to monocytes were minimal. Among CD4^+^ and CD8^+^ lymphocytes, cefuroxime increased IFN-γ and reduced viability (at low and high doses) and increased IL-2 and IL-2R at higher doses. Among CD4^+^ lymphocytes, at both doses, cefuroxime increased %T_h1_ population and decreased %T_h2_, %T_h17_, IL-17A, FOX-P3, and T-bet expression. Among the T_h1_ population, changes were only seen at higher cefuroxime doses, including increased viability and PD-1 and a decrease in FAS, IFN-γ and CD28, and IL-7R.

These findings all form part of the T_h1_ lymphocyte subset response, strongly suggesting that the immunomodulatory mechanism of action of cefuroxime is on this specific lymphocyte subset.

Lymphocytes are integral to the adaptive immune response. In response to the surgical insult, they activate, proliferate, and differentiate into subsets each with specific functions. Following resolution of inflammation, they undergo a natural process of apoptosis, leaving memory cells which are capable of reactivating upon repeated exposure to the same stimulus. Too few of these cells, lymphopenia, and impairments in function of the remaining cells, anergy, are associated with an increased risk of developing post-operative infections [8,20,21]. Maintaining both the number and function of lymphocytes in the immediate post-operative period could reduce the incidence of post-operative infections.

CD4^+^ lymphocytes are responsible for the elimination of intra- and extracellular pathogens, and clearance of tumours which evade cytotoxic CD8^+^ lymphocytes [22,23] and are therefore critical in the immediate post-operative period to prevent both infections and tumour recurrence in cancer surgery. IL-2 and its receptor are important in the proliferation and differentiation of CD4^+^ lymphocytes, specifically of naïve to helper T-cell subtypes following infection [24]. Additionally, IL-2 promotes proinflammatory cytokine release from helper T-cells aiding the initial immune response. However, prolonged release induces an immunosuppressive phenotype by inhibiting cytokine release in T_h17_ cells, a subset especially important in bacterial clearance, and has additional immunoregulatory and suppressive effects on the T_reg_ subset, which have broadly immunosuppressive effects important in the resolution of inflammation. Post-operatively, IL-2 concentration and IL-2R expression decrease, with a nadir around days 3–4 post-operatively, which corresponds with the timing of the development of post-operative infections in our cohort [25,26,27]. Whilst this suggests that reduced IL-2 and its receptor may be preventing CD4^+^ proliferation and differentiation, therefore increasing the risk of post-operative infections, administration of therapeutic IL-2 following surgery [28] did not prevent the initial lymphopenia. Strategies which ameliorate acute reductions in IL-2 cytokine concentration and receptor expression, however, may be beneficial.

We demonstrate that cefuroxime was able to increase both intracellular IL-2 and surface IL-2R expression. Cephalosporins, including cefuroxime, have previously been shown to have immunomodulatory effects on CD4^+^ lymphocyte proliferation, although at high doses they may cause downregulation of IL-2R gene expression and inhibition of proliferation [29,30,31,32,33]. This effect has not been reliably reproduced with beta-lactams, despite the relative cross-reactivity between the two classes [33] suggesting this may be a cephalosporin-only effect.

Whilst this suggests that using cefuroxime as prophylaxis could have beneficial effects, excessive IL-2R activation is also associated with increased cell death through IL-2 activation-induced cell death. Several mechanisms are implicated, including the Fas receptor, mitochondrial, and caspase-3 mediated cell death pathways [34,35]. Beta-lactams induce cell death via direct damage to cellular DNA and through mitochondrial pathways [36,37]. Whilst the assessed antibiotics reduced overall cell viability, cefuroxime was associated with increased viability and a reduction in Fas expression in the T_h1_ subset, suggesting a potential protective effect in this subset. T_h1_ lymphocytes are important in the immune response against intracellular pathogens and activate antigen-presenting cells, including monocytes. However, it should be noted that there was an increase in PD-1, a receptor also implicated in T-cell anergy. Cephalosporins upregulate T_h1_ cell PD-1 expression in a mouse model of pneumonia; however, the association of antibiotics with this receptor and eventual cell death remains unexplored [38]. This suggests that the potential beneficial effects of using cephalosporins on normalisation of IL-2 and IL-2R may be countered by their effects on enhancing other immunosuppressive pathways. The balance of immunostimulatory and immunosuppressive effects in vivo, therefore, warrants further investigation.

We show that cefuroxime modulates the T_h1_/T_h2_ balance by enhancing the proportion of the T_h1_ subset, whilst reducing proportions of the T_h2_, T_h17_, and T_reg_ subsets. Our findings are similar to previous data demonstrating that cephalosporins (ceftriaxone rather than cefuroxime) reduce the proportion of T_regs_ in a mouse model [39], whilst cefuroxime has previously been shown to downregulate genes associated with T_h2_ and T_reg_ subset differentiation in PBMCs [33]. This was not a beta-lactam class effect, as ampicillin increased expression of these genes, suggesting a cephalosporin-specific effect. This is reinforced by the lack of effect of amoxicillin on markers associated with differentiation in our study, although we did not perform in-depth characterisation with amoxicillin or metronidazole. Whilst T_h1_ cells are important in the immune response to clear intracellular pathogens, both T_h2_ and T_h17_ cells aid in extracellular pathogen clearance, the latter especially at mucosal and epithelial barriers, which are often breached during surgery, and are therefore important in preventing surgical site infections. The clinical benefit of increasing the T_h1_ lymphocyte subpopulation whilst reducing the others is unknown.

In sepsis, T_regs_ have an immunosuppressive role [40], although their role in the peri-operative immune response is less clear. An increase in the proportion of this subset occurs following surgery [41]. This is associated with an increased risk of tumour recurrence [42,43], but not with the development of infectious complications [44]. Conversely, low pre-operative T_regs_ levels have been associated with the risk of developing cardiovascular complications in the post-operative period in patients undergoing non-cardiac surgery, although post-operative counts were not predictive [45]. It is therefore unclear whether the reduction in the T_reg_ subset proportion induced by cefuroxime has any clinically relevant effect on peri-operative complications.

IFN-γ produced by CD4^+^ lymphocytes has important roles in tissue homeostasis, immune and inflammatory responses, and tumour immunosurveillance [46]. It acts on innate immune cells including macrophages, enhancing proinflammatory functions, and has local paracrine effects on other CD4^+^ lymphocytes, promoting T_h1_ function but suppressing T_h17_ response. Preliminary clinical studies of peri-operative IFN-γ administration have shown features consistent with an improvement in immune cell function, including increased monocyte HLA-DR expression and enhanced T_h1_ reactivity to specific antigens, but have yet to show clinical benefit [47,48]. We demonstrated that cefuroxime caused an increase in intracellular IFN-γ concentration, and this drives differentiation into T_h1_ cells. However it should be noted that high cefuroxime doses may impair IFN-γ release [49]. This may explain why IFN-γ concentration was reduced specifically in the T_h1_ subset and why the percentage of the T_h17_ subset was reduced at higher cefuroxime concentrations. Despite changes in intracellular IFN-γ concentration, we observed no differences in released cytokines, therefore the clinical effect of increased IFN-γ on enhancing the immune response may be limited, although cytokine release was measured from total PBMCs, rather than in isolated lymphocyte subsets.

Different classes of beta-lactams have heterogenous effects on the immune system [33,50]. This is reflected by the difference in allergy profiles to the different beta-lactams, which is likely due to the presence of only some cross-reactivity between amoxicillin and cephalosporins [51]. Therefore, immunomodulatory effects may be limited to specific antibiotic classes’ potentially supporting their preferential use over others.

Despite the potential beneficial immunomodulatory effects of cefuroxime demonstrated *ex vivo*, a clinical effect was not demonstrated in our cohort. This could be due to our limited sample size; the immune response to surgery is itself a large driver of immunophenotypical changes and therefore could mask the comparatively smaller additional effect of antibiotics. Additionally, alterations to lymphocyte phenotype induced by antibiotics may not alter the risk of post-operative complications following surgery [44]. Our results should be regarded as exploratory, warranting further investigation.

We chose to assess the immune response in the immediate post-operative period; results from patients pre-operatively and at later stages in their recovery may have yielded different findings. Our results are confounded by the high prevalence of active cancer and neoadjuvant chemotherapy in our cohort, and use of intra-operative dexamethasone [52]. Additionally, our patients all received peri-operative antibiotics prior to blood sampling for our in vitro analysis. All in vitro experiments were performed using a single concentration and strain of HKB or CD3/CD28 beads; a different immunological effect may have been seen with other concentrations or stimuli. However, as immune cells are stimulated by the immediate surgical response, the additional effect of another stimulus was minimal. Additional effects of antibiotics on stimulated cell functions may have become apparent at later time points during the patient recovery period, but as antibiotics were not administered beyond the immediate peri-operative period, the clinical relevance would be reduced. Experiments were performed on PBMCs rather than isolated cell populations, resulting in cell signalling between different cell subsets. We were also unable to account for differences in antimicrobial cover provided by the different antibiotics [14]. Our findings cannot be extrapolated to different patient populations and clinical practice.

## 4. Materials and Methods

### 4.1. Ethics

Ethical approval for obtaining clinical samples and data was received from the London—Queen Square Research Ethics Committee (REC reference 20/LO/1024) on 20 October 2020. Informed consent was obtained from all subjects involved in the study.

### 4.2. Clinical Study Participants

We conducted a prospective observational study of patients aged ≥ 18 years who were undergoing major elective surgery at University College London (UCL) Hospitals between 1 August 2021 and 31 July 2022. This was a sub-study of a previously published study investigating the early changes to immunophenotype on subsequent infections in high-risk patients undergoing non-cardiac major surgery [8]. In this study, preliminary analysis of changes to immune cell phenotype on exposure to antibiotics *ex vivo* was conducted in a cohort of 12 patients followed by an in-depth characterisation of immune cell phenotype in a cohort of 16 patients.

Major surgery was defined as a requirement for planned admission to the post-anaesthetic care unit (PACU) [53]. Routinely collected patient data, including demographics, clinical data (physiology, diagnoses), laboratory data, and clinical outcomes, were obtained from electronic healthcare records. The post-operative mortality risk was calculated using the surgical outcome risk tool (SORT) score [54], and the presence of a post-operative infection defined by the standardised endpoints in peri-operative medicine—core outcome measures for peri-operative and anaesthetic care (StEP-COMPAC) criteria [55]. Patients were followed up to hospital discharge or death.

### 4.3. Sample Processing

Patients were recruited prior to surgery. On arrival to the PACU, venesection was performed and 8 mls of blood drawn into a cell preparation tube with sodium heparin (CPT^TM^) vacutainer and a further 5 mL drawn into a serum tube (Beckton Dickinson, Wokingham, UK). Samples were processed within 1 h of venesection. CPT^TM^ vacutainers were centrifuged at 1500 g for 15 min at room temperature and the peripheral blood mononuclear cell (PBMC) layer extracted, washed twice in 2 mls of phosphate-buffered saline (PBS) before being resuspended in freezing media (foetal bovine serum (Gibco, Thermo Fisher Scientific, Paisley, UK) with 10% dimethyl sulfoxide (DMSO, Merck Life Sciences, Gillingham, Dorset, UK)) frozen to −80 °C using isopropyl alcohol (Mr Frosty^TM^, Thermo Fisher Scientific, Paisley, UK), and transferred to liquid nitrogen within 48 h for long term storage.

Samples were analysed in batches. Frozen PBMCs were defrosted by resuspension in RPMI Glutamax medium (Gibco, Thermo Fisher Scientific, Paisley, UK) with 10% foetal bovine serum (FBS) (Thermo Fisher Scientific, Paisley, UK), washed once in media, counted (Countess 3 Automated cell counter, Thermo Fisher Scientific, Paisley, UK), and diluted to a concentration of 1 × 10^6^/mL.

Cell types were chosen based on the existing literature and preliminary experiments [8]. (Supplemental Methods).

### 4.4. Antibiotic Stimulation

Patient PBMCs (1 × 10^6^/mL) were plated into 96-well plates and incubated at 37 °C, 5% CO_2_ for 24 h (for monocyte analysis), or 72 h (for lymphocyte analysis) with or without antibiotics. Amoxicillin (Wockhardt, Wrexham, UK) and cefuroxime (Flynn Pharma, Stevenage, UK) were first resuspended in distilled water as per manufacturer’s instructions. Metronidazole (B. Braun Medical, Sheffield, UK), which came pre-dissolved, and the prepared amoxicillin and cefuroxime stocks were then diluted to working stock concentration in phosphate-buffered saline. Final concentrations of 5 µg/mL and 25 µg/mL for all antibiotics were chosen to represent lower and higher mean serum concentrations, respectively, based on data from pharmacokinetic data obtained in critically ill patients and patients undergoing major surgery [56,57,58,59].

To model the effect of antibiotics on PBMC response to a post-operative infection, patient PBMCs were additionally incubated with either heat-killed *E. coli* 0111:B4 (HKB, Thermo Fisher Scientific, Paisley, UK) at a concentration of 10^8^/mL for 24 h (for monocyte analysis) or CD3/CD28 beads (Miltenyi Biotec, Woking, UK) at a concentration of 4:1 (beads: PBMCs) for 72 h (for lymphocyte analysis) with or without antibiotics. The dose of HKB and CD3/CD28 beads used for experiments in this study were ascertained in recent dose-finding experiments [8,60]. (Supplemental Methods).

Following incubation, plates were centrifuged at 400 g for 5 min at room temperature and the supernatant removed. To assess anergy in the in-depth lymphocyte characterisation panel, the plates were centrifuged as described every 24 h, and the supernatant removed and stored frozen for subsequent cytokine ELISA measurement and the cell pellet then resuspended in media with or without antibiotics as required.

### 4.5. Flow Cytometry

The cell surface markers and intracellular proteins analysed reflected pathways typically associated with sepsis and peri-operative immunosuppression [3,8,9,61]. To assess the effect of antibiotics on markers associated with monocyte phenotype, PBMCs were resuspended in cell staining buffer (Biolegend, London, UK) and incubated with fluorochrome-labelled antibodies to the following markers (CD14, CD16, HLA-DR, CD80, CD86, and CD274 (PD-L1)), viability stain (Aqua UV Live/Dead), and, after use of the Cytofix/perm fixation/permeabilization kit (Beckton Dickinson, Wokingham, UK) as per manufacturer recommendations, fluorochrome-labelled antibodies to the following intracellular cytokines (IL-1β, IL-6, IL-10, and TNF-α).

To assess the basic phenotype of lymphocytes, PBMCs were resuspended in cell staining buffer with fluorochrome-labelled antibodies to the following markers (CD3, CD4, CD8, CD19, CD25 (IL-2RA), CD28, CD127 (IL-7RA), CD152 (CTLA-4), and CD279 (PD-1)), viability stain (Aqua UV Live/Dead), and, after fixation/permeabilization, fluorochrome-labelled antibodies to the following intracellular cytokines (IL-2, IL-10, and IFN-γ). Details of products and concentrations used are listed in Appendix A.

Based on preliminary results, we performed a more focused and in-depth characterisation on T- lymphocytes. PBMCs were resuspended in cell staining buffer with fluorochrome-labelled antibodies to the following cell surface markers (CD3, CD4, CD8, CD19, CD25 (IL-2RA), CD28, CD95 (Fas), CD127 (IL-7RA), CD152 (CTLA-4), CD194 (CCR4), CD196 (CCR6), CD274 (PD-L1), CD279 (PD-1), HLA-DR), viability stain (Zombie near-infrared Live/Dead), and, after use of the Tru-nuclear fixation/permeabilization kit (Biolegend, London, UK) as per manufacturer recommendations, fluorochrome-labelled antibodies to the following intracellular cytokines (IL-2, IL-4, IL-10, IL-17A, and IFN-γ), proteins (NF-κB), and transcription factors (Fox-P3, STAT5, and T-bet). Details of products and concentrations used are listed in Appendix A.

For statistical analysis, lymphocyte differentiation was assessed using the following variables: CCR4, CCR6, IL-4, IL-17A, T-bet, STAT5 (all MFI), with Fox-P3 (MFI), T_h1_, T_h2_, T_h17_, and T_reg_ (all % of parent population) for CD4^+^ lymphocytes, or T_c1_, T_c2_, and T_c17_ (all % of parent population) for CD8^+^ lymphocytes. To assess T_h1_ subset phenotype, the following variables were included: CD28, Fas, IL-10, PD-1, IL-2R, IL-2, HLA-DR, IFN-γ, PD-L1, CTLA-4, IL-7R, NF-κB (all MFI), divided, and viability (both %).

Cells were acquired on an ID7000 spectral cell analyser (Sony Biotechnology Inc, Weybridge, UK) and analysed using ID7000 software (version 1.2, Sony Biotechnology Inc, Weybridge, UK). Alignment check beads (Sony Biotechnology Inc, Weybridge, UK) were used prior to running each experiment, and spectral references for each fluorochrome were added using either single stain labelled heat-killed cells (60 °C for 10 min, live/dead stain) or compensation beads (Beckton Dickinson, Wokingham, UK, all other markers) with appropriate negative controls. FMO (fluorescence minus one) samples were used to identify cell populations. The stopping gate was set at 10,000 events for either CD14^++^CD16^−^ monocytes or CD4^+^ lymphocytes. An example gating strategy is shown in Appendix A.

### 4.6. Cytokine ELISA

Supernatant IFN-γ and IL-10 were measured using Duoset ELISA kits (R&D Systems, Abingdon, UK) as per manufacturer instructions. Samples were diluted 1:2 in reagent dilutant. Optical densities were acquired on a SPECTROstar Nano microplate reader (BMG Labtech, Aylesbury, UK).

### 4.7. Statistics

Clinical and demographic data are presented either as median (inter-quartile range) or number (percentage). Flow cytometry data are presented as either median fluorescence intensity (MFI; arbitrary units), percentage positive cells, percentage of parent population, and compared using Friedmans test without post hoc correction or Wilcoxon test for multiple or paired analyses, respectively.

To identify statistically significant discriminators between two groups, we conducted multiple comparisons using a Mann–Whitney test and calculated a corrected *p*-value (−log10) with a False Discovery Rate of 5%, and data presented using a volcano plot. Time-to-event analysis was conducted using log-rank with a Gehan–Breslow–Wilcoxon test and presented as a Kaplan–Meier plot. Graphs were constructed, and statistical analysis performed using Prism (version 10, GraphPad, San Diego, CA, USA).

## 5. Conclusions

In summary the choice of antibiotics impacts immune function *ex vivo* following major surgery, with cefuroxime associated with immunomodulatory effects on CD4^+^ lymphocytes. The functional implications on the development of subsequent post-operative infectious complications and long-term cancer-free survival require further investigation.

## Figures and Tables

**Figure 1 antibiotics-14-01026-f001:**
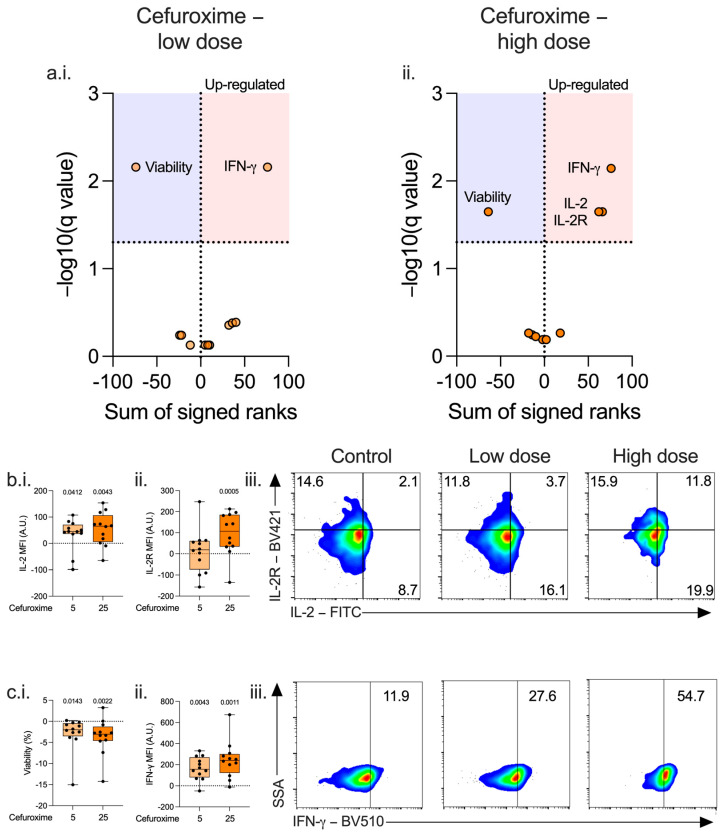
Effect of cefuroxime on lymphocyte immunophenotype. PBMCs isolated from patients immediately post-operatively (n = 12) were incubated for 72 h alone or with low (5 μg/mL, (**a.i.**), light orange) or high (25 μg/mL, (**a.ii.**), dark orange) doses of cefuroxime and the effect on CD4^+^ lymphocyte immunophenotype assessed. Data expressed as volcano plots generated by calculating a corrected q-value (−log10) using a False Discovery Rate (FDR) of 5% using the two-stage step-up method of Benjamini, Krieger, and Yekutieli. Red area represents markers upregulated by cefuroxime compared to control, blue area those markers downregulated. A dose-dependent effect was seen on IL-2 concentration (**b.i.**,**b.iii.**), IL-2R expression (**b.ii.**,**b.iii.**), viability (**c.i.**), and IFN-γ concentration (**c.ii.**,**c.iii.**). Data expressed as box and whisker plots normalised to control; dot represents the individual patient, horizontal line the median, box the interquartile range, and whisker the range, compared using Friedmans test (only *p* < 0.05 show) or as contour plots, where numbers represent percentage.

**Figure 2 antibiotics-14-01026-f002:**
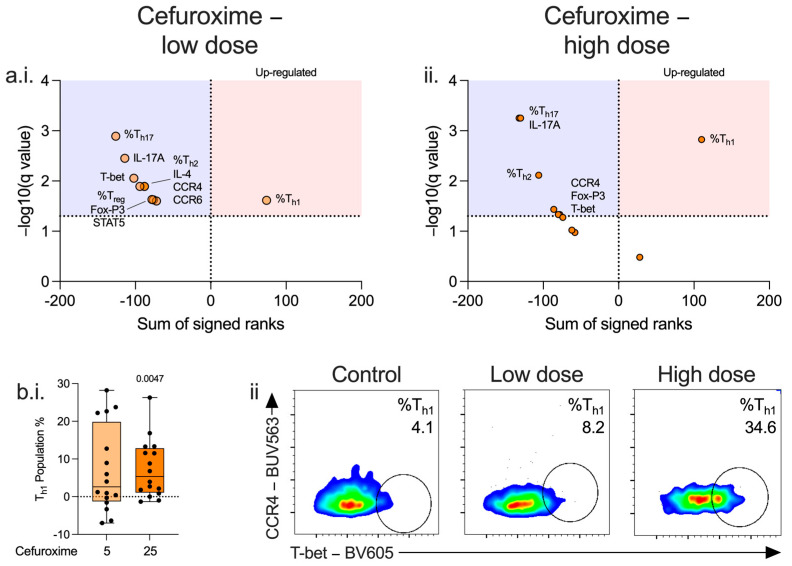
Effect of cefuroxime on CD4^+^ lymphocyte differentiation. PBMCs isolated from patients immediately post-operatively (n = 16) were incubated for 72 h alone or with low (5 μg/mL, (**a.i.**), light orange) or high (25 μg/mL, (**a.ii.**), dark orange) doses of cefuroxime and the effect on CD4^+^ lymphocyte differentiation (**a.**,**b.**) was assessed. Data expressed as volcano plots generated by calculating a corrected q-value (−log10) using a False Discovery Rate (FDR) of 5% using the two-stage step-up method of Benjamini, Krieger, and Yekutieli. Red area represents markers upregulated by cefuroxime compared to control, and blue area those markers downregulated. A dose-dependent effect was seen on the T_h1_ population differentiation (**b.i.**). Data expressed as box and whisker plots normalised to control, dot represents the individual patient, horizontal line the median, box the interquartile range, and whisker the range, compared using Friedmans test (only *p* < 0.05 show) or as contour plots, where numbers represent percentage (**b.ii.**).

**Figure 3 antibiotics-14-01026-f003:**
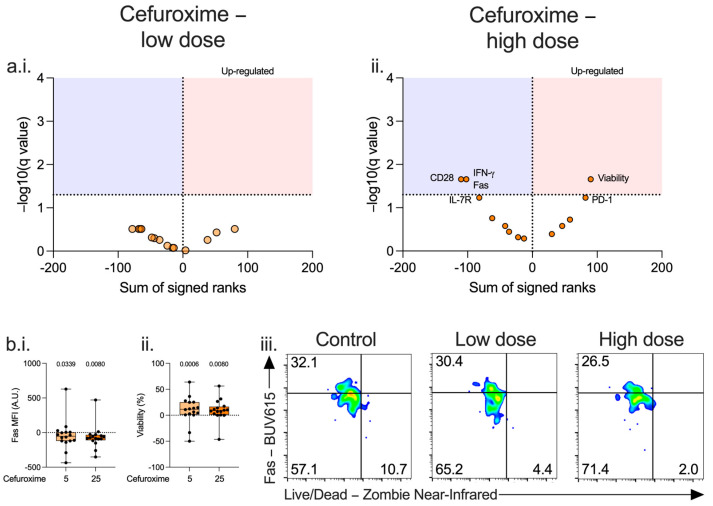
Effect of cefuroxime on T_h1_ lymphocyte immunophenotype. PBMCs isolated from patients immediately post-operatively (n = 16) were incubated for 72 h alone or with low (5 μg/mL, (**a.i.**), light orange) or high (25 μg/mL, (**a.ii.**), dark orange) doses of cefuroxime and the effect on T_h1_ lymphocyte immunophenotype (**a.**,**b.**) assessed. Data expressed as volcano plots generated by calculating a corrected q-value (−log10) using a False Discovery Rate (FDR) of 5% using the two-stage step-up method of Benjamini, Krieger and Yekutieli. Red area represents markers upregulated by cefuroxime compared to control, blue area those markers downregulated. A dose-dependent effect was seen on the T_h1_ population Fas receptor expression (**b.i.**) and viability (**b.ii.**). Data expressed as box and whisker plots normalised to control, dot represents the individual patient, horizontal line the median, box the interquartile range, and whisker the range, compared using Friedmans test (only *p* < 0.05 show) or as contour plots, where numbers represent percentage (**b.iii.**).

**Table 1 antibiotics-14-01026-t001:** Clinical characteristics of antibiotic characterisation cohort.

Variable		Antibiotic Characterisation Cohort (n = 28)	Immunophenotyping Cohort (n = 12)	Lymphocyte Characterisation Cohort (n = 16)
Age (years)		64 (54–75)	61 (50–70)	64 (57–77)
Biological Sex (% male)		15 (54%)	8 (67%)	7 (44%)
BMI		25 (21–30)	24 (21–27)	26 (22–32)
Co-morbidities				
	Hypertension (%)	13 (46%)	7 (58%)	6 (38%)
	Cardiovascular disease (%)	5 (18%)	3 (25%)	2 (13%)
	Respiratory disease (%)	6 (21%)	3 (25%)	3 (19%)
	Type 2 diabetes (%)	6 (21%)	4 (33%)	2 (13%)
	ASA Grade (%)	3 (3–3)	3 (3–3)	3 (3–3)
	Active cancer (%)	27 (96%)	12 (100%)	15 (94%)
	Cancer staging	2 (2–3)	2 (2–2)	2 (2–2)
	Neoadjuvant chemotherapy (%)	17 (61%)	8 (67%)	9 (56%)
SORT Score (%)		1.48 (0.0–2.1)	1.5 (1.4–3.2)	1.1 (0.7–1.6%)
Type of surgery				
	Upper GI (%)	16 (57%)	9 (75%)	7 (44%)
	Lower GI (%)	2 (7%)	0%	2 (13%)
	Maxillofacial (%)	5 (18%)	2 (17%)	2 (13%)
	Gynaecological (%)	5 (18%)	1 (8%)	4 (25%)
Peri-operative antibiotics				
	Prophylaxis administered (%)	28 (100%)	12 (100%)	16 (100%)
	Duration of prophylaxis (days)	1 (1–1)	1 (1–1)	1 (1–1)
	Cefuroxime and metronidazole	17 (61%)	9 (75%)	8 (50%)
	Co-amoxiclav	5 (18%)	2 (17%)	3 (19%)
	Co-amoxiclav and gentamicin	1 (4%)	0%	1 (6%)
	Co-amoxiclav and teicoplanin	1 (4%)	0%	1 (6%)
	Ciprofloxacin and clindamycin	2 (7%)	1 (8%)	1 (6%)
	Ciprofloxacin and metronidazole	1 (4%)	0%	1 (6%)
	Gentamicin	1 (4%)	0%	1 (6%)
Intra-operative dexamethasone use (%)		23 (82%)	11 (92%)	12 (75%)
Operation duration (mins)		302 (220–409)	247 (211–380)	328 (252–446)
Blood loss (mls)		300 (200–600)	200 (200–625)	300 (200–525)
Peri-operative blood transfusion (%)		2 (7%)	1 (8%)	3 (19%)
Post-op infection		11 (39%)	4 (33%)	7 (44%)
	Chest	7 (25%)	2 (17%)	5 (31%)
	Wound	3 (11%)	2 (17%)	1 (6%)
	Other/Unclear	2 (7%)	0%	2 (13%)
Clavien–Dindo classification		2 (1–2)	2 (2–2)	2 (1–2)
Hospital length of stay (days)		12 (8–20)	10 (8–13)	13 (8–23)
In-hospital death (%)		2 (7%)	2 (9%)	0%

Abbreviations: ASA: American association of anaesthesiologists, GI: gastrointestinal.

## Data Availability

The raw data supporting the conclusions of this article will be made available by the authors on request without undue restriction.

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
