# Peer review of "Peri-Operative Antimicrobial Prophylaxis Modulates CD4^+^ Lymphocyte Immunophenotype *Ex Vivo* in High-Risk Patients Undergoing Major Elective Surgery—A Preliminary Observational Study"

_antibiotics, 2025, doi:10.3390/antibiotics14101026_

Round 1
Reviewer 1 Report
Comments and Suggestions for Authors
The study is interesting but are preliminary reports. (preliminary study). You could need to clarify in the title
The most of patient underwent oncological surgeries, (96%) for cancer resection. It would a bias? Oncological o not oncological group will need to be compared?
Author Response
The study is interesting but are preliminary reports. (preliminary study). You could need to clarify in the title
We have amended the title to the following based on comments by yourself and other reviewers “Peri-operative antimicrobial prophylaxis modulates CD4+ lymphocyte immunophenotype ex vivo in high-risk patients undergoing major elective surgery – A preliminary observational study”.
The most of patient underwent oncological surgeries, (96%) for cancer resection. It would a bias? Oncological o not oncological group will need to be compared?
We agree with your comment and had included the following statement in our limitations section “Our results are confounded by the high prevalence of active cancer and neoadjuvant chemotherapy in our cohort”.
Given the small number of patients who did not have active cancer, a meaningful analysis of cancer vs non-cancer would unfortunately not be possible in our cohort. We hope other group with non-cancer patient cohorts will be able to replicate our findings, hence our statement “Our results should be regarded as exploratory, warranting further investigation”.
Reviewer 2 Report
Comments and Suggestions for Authors
Authors analyzed the ex vivo immunomodulatory properties of commonly used antibiotics (amoxicillin, cefuroxime, metronidazole, or combined cefuroxime-metronidazole) on monocyte and lymphocyte phenotype in patients undergoing major elective surgery. This study has functional implications on the better understanding the development of subsequent post-operative infectious complications and choice of cefuroxime in preoperative surgical prophylaxis. I suggest that manuscript can be accept in present form.
Author Response
Thank you for your comments.
Reviewer 3 Report
Comments and Suggestions for Authors
In the manuscript by S. Paketci and colleagues, the authors ask whether commonly used peri-operative antibiotics modulate the immediate post-operative immune state using ex vivo PBMC assays. While the clinical questions are interesting, the readability and quality of the manuscript needs to be improved.
- The reviewer thinks the current title is too narrow and does not capture the central objective (peri-operative antibiotics as immunomodulators of post-op PBMC phenotypes).
- Main and Supplementary figures are too low-resolution to read. Please provide ≥300 dpi images with legible fonts and panel legends. The invisibility created some issues when interpreting the results.
- In the introduction, the reviewer suggests authors to clarify the “40%”, specifically on the procedure types and whether this is all post-operative infections or SSI?
- Some wordings can be considered to be changed. For example, Liberal usage -> routine peri-operative prophylaxis, microbial cover -> antimicrobial spectrum.
- The reviewer asks the authors to explain why amoxicillin, cefuroxime, and metronidazole were selected. Note why other common classes (e.g., penicillin) weren’t pursued.
- Line 64-67, the reviewer suggests the authors to do a thorough literature review on the current knowledge regarding the antimicrobial prophylaxis on post-operative immunosuppression and what is the missing puzzle? This can better explain the significance of the study.
- The reviewer suggests the authors to give percentages with numerators (e.g., 27/28 (96%) cancer).
- In table 1, many regimens show 0% in the 28-patient subset. The reviewer suggests that either (i) collapse to the three actually used (cefuroxime+metronidazole; co-amoxiclav; ciprofloxacin+clindamycin) and move the rest to Supplement, or (ii) retain all rows but footnote that they reflect institutional options and that ex vivo testing was independent of the clinical regimen.
- Figure 1bi/bii, boxes look similar (applies to other figures), yet p-values are small. The reviewer suggests show within-patient shifts in supplementary (spaghetti/slope plots or normalized ΔMFI vs control) and report median paired Δ (95% CI), such that readers can see the paired effect driving the significance.
- In all main figures, the authors did not explain or cite each sub-plot in the main text. More explanation, in-text citation, and data interpretation are required. For example, the whole section the talks about Figure 1 is from line 87 to 91. Please explicitly reference and interpret each sub-panel (1ai 1aii 1bi 1bii 1biii 1ci 1cii 1ciii) in the main text.
- The reviewer asks the authors to justify the concentration: HKB coli 10^8/mL and CD3/CD28 4:1 (source/optimization). The reviewer cannot find how these concentrations/ratios were chosen or optimized (manufacturer guidance, literature, or pilot titration). Please add a rationale and, if available, titration data (dose–response for cytokines/viability) showing you’re operating in a submaximal, dynamic range.
- Line 199, the authors cited a previous paper and mentioned “specific changes”. This is too vague and the reviewer suggest the authors to summarize the changes for the readers.
- Line 202, Open with context, then the main finding (don’t lead with conclusions).
- The reviewer suggests the authors to improve the narrative flow of the discussion to improve the logic, readability, and key take home messages.
Author Response
- The reviewer thinks the current title is too narrow and does not capture the central objective (peri-operative antibiotics as immunomodulators of post-op PBMC phenotypes).
We have amended the title to the following based on comments by yourself and other reviewers “Peri-operative antimicrobial prophylaxis modulates CD4+ lymphocyte immunophenotype ex vivo in high-risk patients undergoing major elective surgery – A preliminary observational study”.
- Main and Supplementary figures are too low-resolution to read. Please provide ≥300 dpi images with legible fonts and panel legends. The invisibility created some issues when interpreting the results.
Our apologies, this appears to have occurred during the .docx to .pdf conversion. This has been rectified.
- In the introduction, the reviewer suggests authors to clarify the “40%”, specifically on the procedure types and whether this is all post-operative infections or SSI?
We have clarified the opening statement to include the request information from the 2 papers referenced: “Post-operative infections (encompassing pneumonia, surgical-site infections, etc) are a significant cause of morbidity, affecting up to 40% of patients undergoing major non-cardiac surgery”.
- Some wordings can be considered to be changed. For example, Liberal usage -> routine peri-operative prophylaxis, microbial cover -> antimicrobial spectrum.
Thank you for the suggestions.
We specifically chose the use of ‘liberal’ given the over-use of antibiotics in the peri-operative population and to reflect the concerted antimicrobial stewardship efforts to reduce inappropriate use. We have added an additional reference to that effect (de Jonge, S.W., et al., Effect of postoperative continuation of antibiotic prophylaxis on the incidence of surgical site infection: a systematic review and meta-analysis. Lancet Infect Dis, 2020. 20(10): p. 1182-1192.)
We have included antimicrobial spectrum as you suggest.
- The reviewer asks the authors to explain why amoxicillin, cefuroxime, and metronidazole were selected. Note why other common classes (e.g., penicillin) weren’t pursued.
To ensure our findings would have maximal impact, we selected the antibiotics most commonly used as antimicrobial prophylaxis worldwide as per a recently published meta-analysis (Fowler, A.J., et al., Liberal or restrictive antimicrobial prophylaxis for surgical site infection: systematic review and meta-analysis of randomised trials. Br J Anaesth, 2022. 129(1): p. 104-113).
To that effect, we included the following statement in our introduction “the beta-lactams (penicillins and cephalosporins) and nitroimidazoles (e.g., metronidazole) being the most commonly used for antimicrobial prophylaxis” and include the above reference.
- Line 64-67, the reviewer suggests the authors to do a thorough literature review on the current knowledge regarding the antimicrobial prophylaxis on post-operative immunosuppression and what is the missing puzzle? This can better explain the significance of the study.
There is currently very little data on the direct immunosuppressive effects of antibiotics in the perioperative patient cohort, this is one of the strengths of our study. We have therefore expanded our sentence in the introduction as follows: “However, there has been considerable effort to reduce the excessive use of antibiotics due to the increasing prevalence of antimicrobial resistance, and an increasing awareness of antibiotic side-effects, including immunosuppression which could paradoxically increase the risk of developing infection [Snow et al – a review of the immunosuppressive effects of antibiotics on immune cells (highlighting the lack of data in patients)]. However, there is currently a paucity of data in the peri-operative patient cohort [Barath et al – an ex vivo study assessing antibiotic effect on peri-operative patient mitochondrial function], therefore the potential effect of antimicrobial prophylaxis on post-operative immunosuppression and clinically relevant outcomes are currently unknown.”
- The reviewer suggests the authors to give percentages with numerators (e.g., 27/28 (96%) cancer).
We have added the numerators to the results section as per your suggestion
- In table 1, many regimens show 0% in the 28-patient subset. The reviewer suggests that either (i) collapse to the three actually used (cefuroxime+metronidazole; co-amoxiclav; ciprofloxacin+clindamycin) and move the rest to Supplement, or (ii) retain all rows but footnote that they reflect institutional options and that ex vivo testing was independent of the clinical regimen.
We have deleted the rows which included 0% as per your suggestion
- Figure 1bi/bii, boxes look similar (applies to other figures), yet p-values are small. The reviewer suggests show within-patient shifts in supplementary (spaghetti/slope plots or normalized ΔMFI vs control) and report median paired Δ (95% CI), such that readers can see the paired effect driving the significance.
We have updated figures 1-3, replacing the box and whisker plots with normalised to control.
Regarding reporting median paired Δ (95% CI), we have previously engaged in vigorous debate with reviewers for prior publications over the best way to present the multitude of differences in different immune markers with different conditions. It has previously been suggested that whilst by including either mean differences or individual p-values for each positive result demonstrates the effect of every individual marker, it reduces the overall readability and flow of the manuscript. As an example, we show below the ‘Effects of antibiotics on post-operative PBMCs in the immunophenotyping cohort’ results section as suggested with median (95%CI). If the reviewer feels this level of detail is important, we will happily rewrite the results section in this style.
“In CD4+ and CD8+ lymphocytes, high-dose amoxicillin increased IFN-g concentration (72 (CI 42 - 212) and 87 (CI -31 - 210) respectively). (Supplemental Figure 2.a.iv. and 2.a.vi) Additionally, in CD8+ lymphocytes, high-dose amoxicillin increased CD28 expression (17 (CI 6 - 90), Supplemental Figure 2.a.vi) with decreased CTLA-4 expression (-60 (CI -252 - -27), Supplemental Figure 2.a.vi) and cell viability (-1.8 (CI -3.8 - -0.4), Supplemental Figure 2.a.vi). CD4+ and CD8+ lymphocyte viability was reduced by metronidazole at low (-3.3 (CI -8.1 - -1.0) and (-4.6 (CI -9.1 - -0.7) respectively, Supplemental Figure 2.c.iii) and high doses (-3.6 (CI -7.4 - -1.3) and -4.3 (CI -13.5 - -1.1) respectively, Supplemental Figure 2.c.iv). Amoxicillin and metronidazole had no effect on monocyte phenotype. (Supplemental Figure 2.a.i., 2.a.ii., 2.c.i., and 2.c.ii.)
In CD4+ and CD8+ lymphocytes, cefuroxime increased IFN-g concentration (157 (CI 69 - 282) and 162 (CI 15 – 238), and 242 (CI 96 - 308) and 207 (CI 104 – 305) at low and high dose respectively, Figure 1.a.i., 1.a.ii., 1.c.ii., and 1.c.iii.) and decreased viability (-2.0 (CI -3.8 - -0.3) and -2.6 (CI -5.0 - -0.1), and -3.1 (CI -4.7 - -0.8) and -4.2 (CI -7.8 - -2.2) respectively, Figure 1.a.i., 1.a.ii., and 1.c.i.) at both doses. There were additional effects at high dose, including an increase in CD4+ lymphocyte IL-2 concentration (65 (CI 2 - 118), Figure 1.b.i. and 1.b.iii.) and IL-2R (107 (CI 33 - 186), Figure 1.b.ii. and 1.b.iii.); and decreased CD8 lymphocyte CTLA-4 (-44 (CI -189 - -1), Supplemental Figure 3.a.v.) and increased IL-2R expression (149 (CI 87 - 200), Supplemental Figure 3.a.v.)). Cefuroxime had no effect on monocyte phenotype. (Supplemental Figure 3.a.i. and 3.a.ii.)
Cefuroxime is often co-administered with metronidazole. When combined with metronidazole, the immunomodulatory effects were similar to that of cefuroxime alone. (Supplemental Figure 3.c) This data suggested cefuroxime may have an immunomodulatory role on lymphocyte function and /or differentiation. We therefore proceeded to evaluate detailed T cell subset phenotype changes induced by cefuroxime.”
- In all main figures, the authors did not explain or cite each sub-plot in the main text. More explanation, in-text citation, and data interpretation are required. For example, the whole section the talks about Figure 1 is from line 87 to 91. Please explicitly reference and interpret each sub-panel (1ai 1aii 1bi 1bii 1biii 1ci 1cii 1ciii) in the main text.
We have done as suggested, an example of which is demonstrated in the response to point 9 above.
- The reviewer asks the authors to justify the concentration: HKB coli 10^8/mL and CD3/CD28 4:1 (source/optimization). The reviewer cannot find how these concentrations/ratios were chosen or optimized (manufacturer guidance, literature, or pilot titration). Please add a rationale and, if available, titration data (dose–response for cytokines/viability) showing you’re operating in a submaximal, dynamic range.
We have added the following sentence to the methods: “The dose of HKB and CD3/CD28 beads used for experiments in this study were ascertained in recent dose-finding experiments (REF: Snow, T.A.C., et al., Early dynamic changes to monocytes following major surgery are associated with subsequent infections. Front Immunol, 2024. 15: p. 1352556.
- Line 199, the authors cited a previous paper and mentioned “specific changes”. This is too vague and the reviewer suggest the authors to summarize the changes for the readers.
We have expended on the sentence as follows; “Surgery induces several changes to the immune system, and we have previously described specific changes associated with post-operative infections including: elevated monocyte cell count, reduced monocyte chemokine receptor expression (CXCR4), and an increase in CD4+ lymphocyte IL-7R expression.”
- Line 202, Open with context, then the main finding (don’t lead with conclusions).
We have removed the conclusion statement regarding Th1/Th2 balance as suggested.
- The reviewer suggests the authors to improve the narrative flow of the discussion to improve the logic, readability, and key take home messages.
We hope the changes made above have improved the flow, please let us know if you have any more specific suggestions.
Round 2
Reviewer 3 Report
Comments and Suggestions for Authors
The reviewer appreciates the authors’ efforts to address prior comments. After careful assessment, the reviewer suggests the following additional modifications:
- References 1–2 do not clearly support “up to 40% infections” nor the restriction to “major non-cardiac surgery.” Please provide infection-specific, procedure-contextual citations, or revise the claim accordingly.
- Antibiotic selection rationale: The Fowler et al. meta-analysis evaluates dosing intensity, not which agents are most commonly used; it does not substantiate choosing amoxicillin, cefuroxime, and metronidazole. Please add guideline/procedure-specific sources that list standard prophylaxis regimens and explain why these agents, and not alternatives, were selected.
- The reviewer appreciates the authors comments and reply regarding Fig 1-3. No need to have detailed version as your reply example. However, to make the paired effect driving small p-values transparent without lengthy text, please add Supplementary spaghetti/slope plots showing within-patient shifts across 0/5/25 for the key endpoints.
- The cited protocol uses 12 h (monocytes) and 48 h (lymphocytes) for HKB E. coli 10⁸/mL and CD3/CD28 4:1, whereas the manuscript uses 24 h and 72 h. Please justify the longer incubation times (e.g., kinetics/assay endpoints) or provide a brief rationale in Methods/Supplement.
- The revised Discussion is largely unchanged in substance, still a recap of results without a coherent through-line to clinical relevance.
Author Response
The reviewer appreciates the authors’ efforts to address prior comments. After careful assessment, the reviewer suggests the following additional modifications:
- References 1–2 do not clearly support “up to 40% infections” nor the restriction to “major non-cardiac surgery.” Please provide infection-specific, procedure-contextual citations, or revise the claim accordingly.
We agree there is a variety in the reporting of the incidence of post-operative infections. Reference 1 which is a global survey encompassing all surgical specialties reports an infection complication rate in Table 2 of 9% although does not stratify the risk by surgical specialty type. In reference 2, data is taken from Table 3: Incidence of postoperative morbidity according to surgical specialty (%). The total percentage (1st column) for any infection (3rd row) is 40.3% this varies from 32.4% for orthopaedic surgery (2nd column) to 64.6% (4th column) in the urology cohort.
We have therefore amended our statement as follows: “Post-operative infections (encompassing pneumonia, surgical-site infections, etc) are a significant cause of morbidity. The reported incidence varies between studies, depending on the definition and reporting of infections, the patient cohort, and type of surgery; ranging from 9% in a global survey of all surgical specialities [ref 1], to 40% of patients undergoing major non-cardiac surgery at a tertiary referral specialist centre [ref 2].”
- Antibiotic selection rationale: The Fowler et al. meta-analysis evaluates dosing intensity, not which agents are most commonly used; it does not substantiate choosing amoxicillin, cefuroxime, and metronidazole. Please add guideline/procedure-specific sources that list standard prophylaxis regimens and explain why these agents, and not alternatives, were selected.
We have further expanded on this section to explain our rationale as follows “The conventional approach to reducing the risk of post-operative infections is the liberal use of antibiotics [10]. However, there has been considerable effort to reduce excessive use of antibiotics due to the increasing prevalence of antimicrobial resistance; guidelines recommend only administrating in cases of clean surgery with prosthesis insertion, or if surgical contamination occurs [11]. Beta-lactams (penicillins and cephalosporins) and nitroimidazoles are the most frequently used classes in a recent meta-analysis of antimicrobial prophylaxis regimens [12] (Supplemental Table 1 – this is a summary table we had generated from the supplemental data included in the meta-analysis by Fowler et al), with amoxicillin, cefuroxime, and/or metronidazole currently recommended by UK guidelines [13]. Cefuroxime and metronidazole may be better in preventing post-operative infections compared to other antibiotics, related to their different antimicrobial spectra [14].
In addition to concerns regarding antimicrobial resistance, understanding and awareness of antibiotic side-effects are increasing. Pre-clinical data demonstrates inadvertent effects of antibiotics on the immune system (REF) [15]. However, it is unclear if this affects patient outcomes, including the risk of late infections [16]. “
- The reviewer appreciates the authors comments and reply regarding Fig 1-3. No need to have detailed version as your reply example. However, to make the paired effect driving small p-values transparent without lengthy text, please add Supplementary spaghetti/slope plots showing within-patient shifts across 0/5/25 for the key endpoints.
We have added and referenced throughout the document an additional Supplemental Figure 3, which includes slope plots of all significant findings in CD4+ lymphocyte population to demonstrate the paired effect.
- The cited protocol uses 12 h (monocytes) and 48 h (lymphocytes) for HKB E. coli 10⁸/mL and CD3/CD28 4:1, whereas the manuscript uses 24 h and 72 h. Please justify the longer incubation times (e.g., kinetics/assay endpoints) or provide a brief rationale in Methods/Supplement.
We have now expanded on our supplemental methods section including adding 3 new supplemental figures to explain the full rationale as follows: We also cite the single time point used as a limitation in our discussion.
“To determine the dose and duration of heat-killed bacteria (E. coli (EC)) stimulus required to elicit an immune response in vitro, healthy volunteer (n=16) whole blood was incubated for 6 hours with or without heat-killed E. coli (EC). The additional effects of antibiotics were also assessed on EC-stimulated cells using flow cytometry. Granulocytes were gated based on forward- and side-scatter, singlets, live and CD66b+, monocytes by HLA-DR positive and then classical monocytes as CD14++CD16-. (Supplemental Figure 11) Ethical approval was granted by the University College London Research Ethics Committee (REC reference 19181/001).
In response to EC-stimulation, there was an increase in markers associated with activation (CD66b) and intracellular cytokine concentration (IL-1b and IL-6) in granulocytes. Few additional effects were demonstrated with antibiotics; high-dose cefuroxime-metronidazole increased intracellular TNF-a concentration. (Supplemental Figure 12)
In response to EC-stimulation, there was an increase in markers associated with chemotaxis (CXCR4 and CCR2), T-cell suppression (PD-L1) antigen presentation (HLA-DR and CD86), and cytokine concentration (IL-1b, IL-6 and TNF-a) in monocytes. When co-incubated with antibiotics; cefuroxime alone and when combined with metronidazole decreased CCR2 expression and high dose cefuroxime reduced HLA-DR expression. (Supplemental Figure 13)
Given the lack of antibiotic effect demonstrated at 6 hours in the healthy volunteer model, we performed subsequent time-course experiments in healthy volunteer PBMCs (n=6) which identified maximal effects on monocyte HLA-DR expression at 24 hours and lymphocyte cell death at 72 hours (at 96 hours there was sharp loss of cell viability). (Supplemental Figure 14) These timepoints were chosen for subsequent experiments on patient PBMCs.”
- The revised Discussion is largely unchanged in substance, still a recap of results without a coherent through-line to clinical relevance.
We have made multiple changes to the discussion to add additional clinical relevance of our main findings where we felt this could be improved.
Round 3
Reviewer 3 Report
Comments and Suggestions for Authors
The reviewer appreciates the author's efforts in addressing the comments. Before being accepted, the authors may included the incomplete or WRONG version of supplementary Figure 14. As you can see, only panel (a) is presented. Please fix this error.
Author Response
The reviewer appreciates the author's efforts in addressing the comments. Before being accepted, the authors may included the incomplete or WRONG version of supplementary Figure 14. As you can see, only panel (a) is presented. Please fix this error.
Thank you for identifying this, the figure was correct but the wrong caption was included. This has now been updated to the following:
Supplemental Figure 14: Time-course of effect of stimulus on monocyte HLA-DR expression and CD4+ lymphocyte viability
Healthy volunteer PBMCs (n=6) were incubated with either (a.i.) heat-killed E coli and the effect on classical monocyte HLA-DR expression assessed at 0, 6, 24 and 36 hours, or (a.ii.) CD3/28 beads and the effect on CD4+ lymphocyte viability assessed at 24, 48, 72 and 96 hours. Data expressed as median fluorescence intensity measured in arbitrary units (MFI (A.U.)) or percentage (%) of population. Individual points represent individual volunteers, horizontal line the median, box the interquartile range and whisker the range.